# A LAMP assay for the rapid and robust assessment of *Wolbachia* infection in *Aedes aegypti* under field and laboratory conditions

**Moshe E. Jasper** *, **Qiong Yang**, **Perran A. Ross**, **Nancy Endersby-Harshman**, **Nicholas Bell**, **Ary A. Hoffmann**

Pest and Environmental Adaptation Research Group, School of BioSciences, The University of Melbourne, Victoria, Australia

* moshe.jasper@unimelb.edu.au

**Data Availability Statement:** All relevant data are within the manuscript and its Supporting Information files.

## Abstract

With *Wolbachia*-based arbovirus control programs being scaled and operationalised around the world, cost effective and reliable detection of *Wolbachia* in field samples and laboratory stocks is essential for quality control. Here we validate a modified loop-mediated isothermal amplification (LAMP) assay for routine scoring of *Wolbachia* in mosquitoes from laboratory cultures and the field, applicable to any setting. We show that this assay is a rapid and robust method for highly sensitive and specific detection of *w*AlbB *Wolbachia* infection within *Aedes aegypti* under a variety of conditions. We test the quantitative nature of the assay by evaluating pooled mixtures of *Wolbachia*-infected and uninfected mosquitoes and show that it is capable of estimating infection frequencies, potentially circumventing the need to perform large-scale individual analysis for *w*AlbB infection status in the course of field monitoring. These results indicate that LAMP assays are useful for routine screening particularly under field conditions away from laboratory facilities.

## Introduction

*Wolbachia* releases are being undertaken in *Aedes aegypti* populations both for replacing existing populations with mosquitoes that have a reduced ability to transmit dengue [1, 2] and other arboviruses, and for suppressing mosquito populations directly due to sterility generated by males infected with *Wolbachia* [3]. A challenge in implementing *Wolbachia*-based strategies is that a high level of quality control is required for release success. This includes ensuring that source mosquito cultures used for releases remain infected by *Wolbachia*. It is also important to track infection status in release areas (particularly with releases aimed at replacing existing populations by *Wolbachia*-infected populations) given that not all releases are successful and periodic interventions may be needed for others [4, 5]. Successful replacement is dependent on *Wolbachia* frequencies in populations exceeding an unstable equilibrium point [1] and potential spread from a release site [6].

*Wolbachia* monitoring has been undertaken with a variety of approaches including staining, electron microscopy and PCR with *Wolbachia*-specific primers. For releases where high

**Funding:** AH received a grant (# 108508) from the Wellcome Trust (https://wellcome.ac.uk/). AH received a grant (Program Grant # 1132412) from the National Health and Medical Research Council (AU), (https://www.nhmrc.gov.au/). AH received a grant (Fellowship Grant # 1118640) from the National Health and Medical Research Council (AU), (https://www.nhmrc.gov.au/). The funders had no role in study design, data collection and analysis, decision to publish, or preparation of the manuscript.

**Competing interests:** The authors have declared that no competing interests exist.

throughput is desirable, *Wolbachia* detection is currently achieved through qPCR fluorescence-based approaches such as RT/HRM (real time PCR/high resolution melt) such as described by Lee et al. [7] which achieves relative quantification of *Wolbachia* on a Roche 480 LightCycler$^{®}$ system allowing samples to be scored in 384-well plates. While this method is efficient and can be used to detect multiple *Wolbachia* strains along with identification of mosquito vectors of disease, it requires expertise and laboratory facilities that are well beyond what is readily available in many developing nations. Each assay also requires a substantial amount of time (around 1.5 h until results are available) as well as a dedicated qPCR machine.

Loop-mediated isothermal amplification (LAMP) is a powerful DNA amplification technique, enabling the detection of trace elements of DNA with high rapidity, sensitivity and accuracy [8]. It involves isothermal amplification through the interaction of four to six primers with up to eight target sites. When combined with a polymerase with high displacement activity, two outer primers assist two compound inner primers to form alternating loop structures within the DNA, providing self-primed single-stranded substrates for further inner primer interaction and replication. This process can be further accelerated by the addition of loop primers also targeted to the exposed single-stranded region of the loop [9].

This technique has found widespread application primarily in the diagnosis and monitoring of infectious diseases such as malaria [10, 11], West Nile virus [12, 13] and dengue [14]. LAMP has also been applied in other contexts including agriculture, quarantine, forensics, and environmental monitoring through DNA [15]. In all these contexts, the LAMP technique allows for fast and accurate assays that can be deployed with equivalent sensitivity to traditional PCR methods, but often with much cheaper costs and less technically-demanding deployment–ideal for field contexts and settings where laboratory expertise is limited.

Three LAMP assays have thus far been published for *Wolbachia* monitoring [16–18]. Gonçalves et al. (2014) targeted *Wolbachia*'s 16S ribosomal protein sequence, and amplified *Wolbachia* across multiple strains, including *w*AlbB, *w*Mel, and *w*MelPop. A study applying this assay to field samples in Malaysia found that it compared favourably with a standard PCR method for *Wolbachia* detection in *Aedes albopictus* mosquitoes, detecting a higher infected rate than observed with PCR [19]. However, it should be noted that this study used an experimental design that did not include the loop primers, which Gonçalves et al. (2014) consider essential to ensure specificity of the assay.

A third *Wolbachia* paper using LAMP [18] independently targeted the 16S ribosomal protein sequence. As in the above *Wolbachia* assay [16], a range of *Wolbachia* strains were targeted and detected in *Ae. albopictus* and *Ae. aegypti*. However, an additional assay was also developed specific to *w*AlbB (ordinarily present within *Ae. albopictus*) and *w*Pip strain *Wolbachia* surface proteins (*wsp*) [17]. This assay originally involved one loop primer and a specific detection method with an additional oligomeric probe designed for presence/absence discrimination of fluorescence by eye. The specific nature of this assay makes it a promising target for adaptation to monitor *Ae. aegypti* populations transinfected with *w*AlbB for disease control.

One important development of LAMP techniques has been to quantitatively assess targets (qLAMP) [20]. These developments have led to stable linear determinations of products across as many as nine orders of magnitude concentration over a wide range of human and agricultural pathogens [20–23]. Such quantitative assessments could be useful for *Wolbachia* monitoring, because a key feature of release success is the frequency of the endosymbiont in field populations. Currently in releases, the *Wolbachia* status of hundreds of mosquitoes is determined at a centralized facility using expensive equipment [1, 6]. In contrast, qLAMP conducted on pools of mosquitoes has the potential to provide rapid and cost-effective estimates of local *Wolbachia* frequencies. When implemented on a device such as the Genie$^{®}$ III,

qLAMP assays require minimal training and are highly portable, reducing the load on centralised monitoring laboratories.

Accordingly, the aims of this research were to (a) adapt and extend the assay of Bhadra et al. [17] using a *w*AlbB *wsp* primer set as well as an *Ae. aegypti ITS1* primer set [24] for efficient and specific detection of *w*AlbB-infected *Ae. aegypti* mosquitoes in the context of control efforts including those in sub-optimal conditions, (b) compare this method to established qPCR monitoring methods on samples taken from field locations, and (c) develop a quantitative form of the assay for use on pooled mosquito extractions to determine relative *w*AlbB frequencies. The results are expected to be applicable to a variety of projects involving *Wolbachia* within health and agricultural contexts.

## Methods

### *Aedes aegypti* samples

**Laboratory colonies.** *Aedes aegypti* were primarily derived from three laboratory colonies: (a) a *w*AlbB-infected colony with a *w*AlbB strain originating from *Ae. albopictus* [25], (b) an uninfected colony originating from wild populations in Cairns, Australia, and (c) a *w*Mel-infected colony with a *w*Mel strain originating from *Drosophila melanogaster* [26]. The three laboratory colonies of *Ae. aegypti* were crossed to mosquitoes of a common Australian background reared in an identical manner to each other, i.e. at 26°C with food provided *ad libitum*. Adults were sacrificed and stored in absolute ethanol before DNA extraction.

**Alternate rearing and storage conditions.** *Aedes aegypti* are exposed to a range of environmental conditions in nature, including heat stress and resource competition, which can produce adults of various sizes and with different *Wolbachia* loads [27]. We altered rearing conditions to simulate several scenarios affecting size and *Wolbachia* density. Small *w*AlbB-infected adults were produced according to Callahan et al. [28] by providing larvae with food *ad libitum* for 3 d and then depriving larvae of food until adulthood. Heat-stressed adults were generated according to Ross et al. [27] by holding eggs at a cyclical temperature regime of 30–40°C for one week. Eggs were then hatched and reared under standard conditions to produce adults with a reduced *Wolbachia* density. Field-collected adults are of variable age and may have taken a blood meal; we therefore tested 30 d old adults and 7 d old females that were stored in absolute ethanol at -20°C either immediately or 24 h after feeding on a human volunteer. Six mosquitoes were tested per treatment for these experiments.

We additionally tested the ability of the LAMP assay to detect *w*AlbB when mosquitoes were stored under suboptimal conditions that may be experienced during field sampling. *w*AlbB-infected adults reared in the laboratory were killed by shaking and stored for 1, 2, 3, 5, 10, 20 and 30 d at 26°C or for 10 d at 37°C in open air before storage in absolute ethanol at -20°C. Dead *w*AlbB-infected adults were also stored in water at 26°C for 3 d before transfer to ethanol and -20°C–simulating adults found floating in ovitraps as part of field collections. Six mosquitoes were tested for each scenario, except for 30 d at 26°C and 10 d at 37°C. As these represent the two most extreme scenarios, we tested 18 mosquitoes for each one.

**Malaysian field samples.** Malaysian samples were collected from three locations (researchers blinded as to origin) by staff from the Institute for Medical Research, Kuala Lumpur (https://www.imr.gov.my), and stored at -20°C in absolute ethanol before extraction. Mosquitoes were collected using ovitraps from two sites in Kuala Lumpur where the release of *w*AlbB-infected mosquitoes with a strain described by Ant et al. [29] is currently underway [2], as well as one control site where mosquitoes with *Wolbachia* have not been released. Mosquitoes were reared under standard conditions and sacrificed as adults for extraction. DNA was extracted from 24 mosquitoes from each field location for further analysis.

## DNA extraction

For most experiments, individual mosquitoes were extracted by placing them in 200 μL of 0.3 M KOH and incubating tubes at 95°C for five minutes. This is the KOH concentration recommended for GeneWorks' Lyse&Lamp reaction buffer for use on the Genie® III.

For pooled extractions, the quantity of KOH was increased–thus, a pool of 99 uninfected and one *w*AlbB-infected *Ae. aegypti* was extracted in 9 mL 0.3 M KOH, with aliquots taken before (i.e. negative control) and after the addition of the lone infected mosquito.

Our standard *Wolbachia* qPCR (LightCycler®) methods do not involve KOH. For comparisons between LAMP (Genie® III) and standard qPCR, genomic DNA was extracted using 250 μL of 5% Chelex® 100 Resin (Bio-Rad laboratories, Hercules CA) and 3 μL of Proteinase K (20 mg/ mL) (Roche Diagnostics Australia Pty. Ltd., Castle Hill New South Wales, Australia) solution. Tubes were incubated for 30 minutes at 65°C then for 10 minutes at 90°C. Following Chelex® extraction, an equivalent volume of 0.6 M KOH (Chem-Supply, Gillman, SA, Australia) was added to aliquots taken from each individual to produce final concentrations of 0.3 M KOH for analyses on the Genie® III, with an unadjusted aliquot being used for the qPCR assay.

## qPCR assays

*Aedes aegypti* from the field sampling were tested for *Wolbachia* infection with qPCR according to Lee et al. [7] via the Roche LightCycler® 480. Three primer sets were used to amplify markers to confirm quality of mosquito DNA, the *Ae. aegypti* species and the presence or absence of the *w*AlbB infection. Crossing point (Cp) values of three consistent replicate runs were averaged to produce the final result. Differences in Cp values between the *Ae. aegypti* and *w*AlbB markers were transformed by $2^n$ to produce relative *Wolbachia* density measures.

## LAMP assays

LAMP primers for the *wAlbB wsp* sequence were derived from Bhadra et al. [17]. However, their OSD probe for *wsp* was replaced by an additional loop primer to increase detection speed (see S1 Table). LAMP primers for the *Ae. aegypti ITS1* gene were taken from Schenkel et al. [24]. Primers were manufactured according to our specifications by Integrated DNA Technologies Inc. (Coralville, IA, USA) under the standard desalting purification process. Two alternative versions of each primer set were prepared to modify speed characteristics–a five primer and a six primer set. The *wsp* 5-primer set was identical to the original Bhadra primers, whereas the *Ae. aegypti ITS1* 5-primer set was constructed by removing the forward loop primer.

Typical LAMP reactions were conducted on a Genie® III machine (OptiGene Limited, Horsham UK)) according to GeneWorks' Lyse&Lamp instructions, using their proprietary ISO-001-LNL Lyse&Lamp buffers. They involved combining 5 μL of a 20-fold dilution of extracted DNA with 20 μL master-mix, itself consisting of 15 μL Lyse&Lamp buffer, as well as enough of each LAMP primer to produce final concentrations of 20 pM FIP & BIP, 10 pM of each loop primer, and 5 pM each F3 & B3 respectively (in a final reaction volume of 25 μL).

Reactions were incubated at 65°C for 20–30 minutes. The Genie® III machine maintains real-time fluorescence detection throughout the incubation. Following amplification, an annealing curve analysis was conducted by reducing temperature by increments from 97 to 78°C in order to confirm the specificity of the amplified products.

## Quantitative validation & frequency curves

To evaluate the quantitative efficacy of the LAMP primers, three *w*AlbB-infected *Ae. aegypti* KOH DNA extractions were first quantified using a Qubit 2.0 fluorometer (Thermo Fisher Scientific), then diluted up to 5,000-fold to form a standard concentration curve for estimating the concentration of the products of the *Ae. aegypti ITS1* and *w*AlbB *wsp* LAMP primer sets. LAMP reactions were then run across this curve for both *w*AlbB and *Ae. aegypti* primer sets, comparing peak amplification time ($T_p$, min) with the log of relative concentration. Following visual inspection, regressions were carried out over the linear sections of that curve using the R function *lm*.

To investigate the ability of qLAMP to detect the relative frequencies of *w*AlbB-infected mosquitoes within a population, pooled DNA mixtures of both *w*AlbB-infected and uninfected individuals were created using laboratory-reared *Ae. aegypti*. Each pool was constructed to contain an equal volume of the DNA extract of twenty mosquitoes of mixed sex, but with different numbers of infected and uninfected mosquitoes. Eight frequency levels were constructed (infected/total): 0/20, 1/20, 3/20, 5/20, 8/20, 11/20, 15/20 20/20, with three separately generated pools at each frequency.

To further investigate the sensitivity of *w*AlbB detection in large pools of individuals where very few are infected, an additional pooled mixture was created by combining a single *w*AlbB-infected individual with 99 uninfected in the same extraction.

LAMP reactions targeting both *w*AlbB *wsp* and *Ae. aegypti ITS1* regions were then run on these pools, using the previously described concentration regressions to derive relative concentrations of *Wolbachia* and *Ae. aegypti* DNA for each pool. The *w*AlbB *wsp* concentrations were then adjusted for overall mosquito DNA concentration based on *Ae. aegypti ITS1* results–i.e. through calculating the ratio of *Wolbachia* concentration to *Ae. aegypti* concentration. Visual inspection and regressions were performed on the resulting frequency estimates, testing for goodness of fit with the known frequencies within each sample.

# Results and discussion

## Primer validation and characterization

The *Ae. aegypti ITS1* primers from Schenkel et al. [24] exhibited a stable annealing point of 92.7˚C (S.D. 0.11˚C). With fresh reagents (i.e. isothermal buffer within two weeks of resuspension important as buffer decline slows amplification times, biasing quantitative assays), the use of all six primers produced positive detections with peak amplification times ($T_p$) ranging between 6 and 12 minutes over a 5,000-fold concentration gradient (S1 Fig).

The adapted *w*AlbB *wsp* LAMP primers from Bhadra et al. [17] (with an additional loop primer, i.e. with 6 primers) successfully amplified DNA from *w*AlbB-infected laboratory mosquitoes up to ten minutes faster than the Bhadra et al. primers alone, with a stable annealing point of 83.9˚C (S.D. 0.17˚C). With fresh reagents, these updated primers produced positive detections with $T_p$ values ranging between 7 and 12 minutes over a 5,000-fold concentration gradient (S1 Fig). This primer was sensitive enough to detect the presence of *w*AlbB-infection in a 5,000-fold dilution of a KOH-extracted DNA from an *Ae. aegypti* individual. We also tested the assay's sensitivity to the presence of infected individuals amongst large pools of uninfected individuals. We could detect a single *w*AlbB-infected *Ae. aegypti* among 99 uninfected mosquitoes with a $T_p$ of 10 minutes–well within the quantitative bounds for this primer set (see S1 Fig).

To investigate the specificity of the modified *w*AlbB *wsp* LAMP primers, we challenged them with *w*Mel-infected *Ae. aegypti*. No amplification was seen over a 30-min period in six *w*Mel-infected samples.

When compared with the original 5-primer *w*AlbB *wsp* LAMP assay [17], the modified 6-primer assay was substantially faster (compare the concentration curve for 5-primer LAMP in Fig 1B with the curve for 6-primer (modified) LAMP in S1 Fig). This is expressed by a five-minute difference in intercept values for the respective regressions. The modified LAMP primer set we have developed thus represents a rapid, specific, and highly sensitive assay for *w*AlbB detection in *Ae. aegypti* mosquitoes.

## Development of qLAMP for *w*AlbB in *Ae. aegypti*

Following KOH extractions, resulting DNA concentrations of three *w*AlbB individuals were first quantified using Qubit, then the samples were diluted up to 5,000-fold to form a standard concentration curve for measuring the *Ae. aegypti ITS1* & *w*AlbB *wsp* LAMP primer sets. Concentration was highly correlated with amplification time under 6-primer conditions for each assay (S1 Fig); however, for highly precise quantitation these reactions were deemed too fast for the 15-second $T_p$ resolution of the current Genie$^®$ III software. Accordingly, we repeated the curves on 5-primer LAMP sets, i.e. where the forward loop primer of each group had been removed (Fig 1).

A regression of 5-primer *Ae. aegypti ITS1* $T_p$ against the natural logarithm of overall DNA concentration was found to be highly significant (p = 1.2 e-13) with an adjusted $R^2$ of 0.918 and a regression coefficient of -1.391 (S.E. 0.086). The regression of the original 5-primer *w*AlbB *wsp* $T_p$ against the natural logarithm of overall DNA concentration was also highly significant (p = 1.09 e-08) with an adjusted $R^2$ of 0.770 and a regression coefficient of -0.8306 (S.E. 0.094). The reduced $R^2$ of *w*AlbB relative to *Ae. aegypti* may partly reflect variable concentrations of *w*AlbB within the three mosquitoes used for the curve.

## Application of qLAMP to pooled DNA

The association of *w*AlbB *wsp* to *Ae. aegypti ITS1* concentration ratios to known frequency of the infection in DNA from pools of 20 mosquitoes is shown in Fig 2. The regression was highly significant (p = 1.10e-06) with a regression coefficient of 0.0377 (S.E. 0.0053) and an adjusted $R^2$ of 0.7098 –higher than the $R^2$ of 0.6091 for a regression of *w*AlbB concentration alone. These patterns suggest that the approach is sufficient for differentiation of *Ae. aegypti* with low, medium, or high *w*AlbB infection frequency.

## Effectiveness under conditions of stress or poor storage

The *w*AlbB infection was readily detected when dead mosquitoes were stored in air for up to 30 d at 26˚C and 10 d at 37˚C–an improvement on the original Bhadra assay, which detected *Wolbachia* in only 40% of individuals stored for a week or more at 4˚C, and failed to detect *Wolbachia* for individuals scored for a week at 37˚C [17]. We obtained rapid amplification times for all storage conditions, though times for adults stored in water for 3 d were somewhat impacted (Fig 3A). Reliable detection of *w*AlbB was also achieved for adults that were aged, blood-fed, heat stressed or nutritionally stressed during development (Fig 3B). The modified LAMP assay we have developed is therefore robust to both poor storage conditions and low titre of *Wolbachia*. These features are valuable for field contexts such as the continued monitoring of *Wolbachia* releases, as the assay will reliably detect infections in low-titre situations and cope well with samples that have undergone degradation before DNA extraction. Our results suggest *w*AlbB will still be detectable where mosquito bodies have dried out in traps for

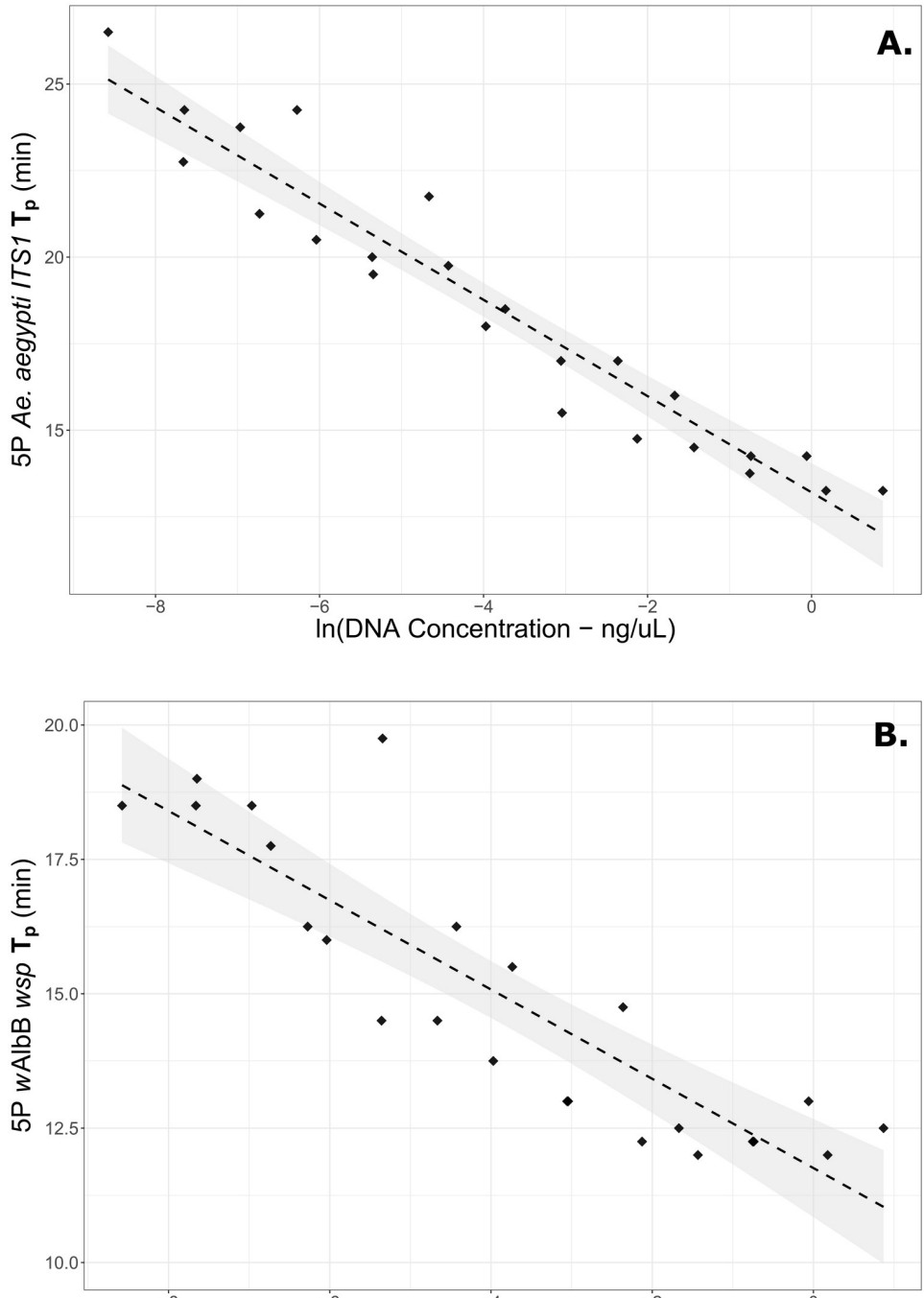

**Fig 1. Concentration curves for (A)** *Ae. aegypti ITS1* **& (B)** *w*AlbB *wsp* **LAMP 5-primer sets (i.e. the forward loop primer of each has been removed).** The horizontal axis shows the natural log of overall extracted DNA in the samples, while the vertical axis shows the peak amplification time ($T_p$) for each primer set under fresh reagent conditions. Linear regression lines are shown with 95% confidence intervals shaded.

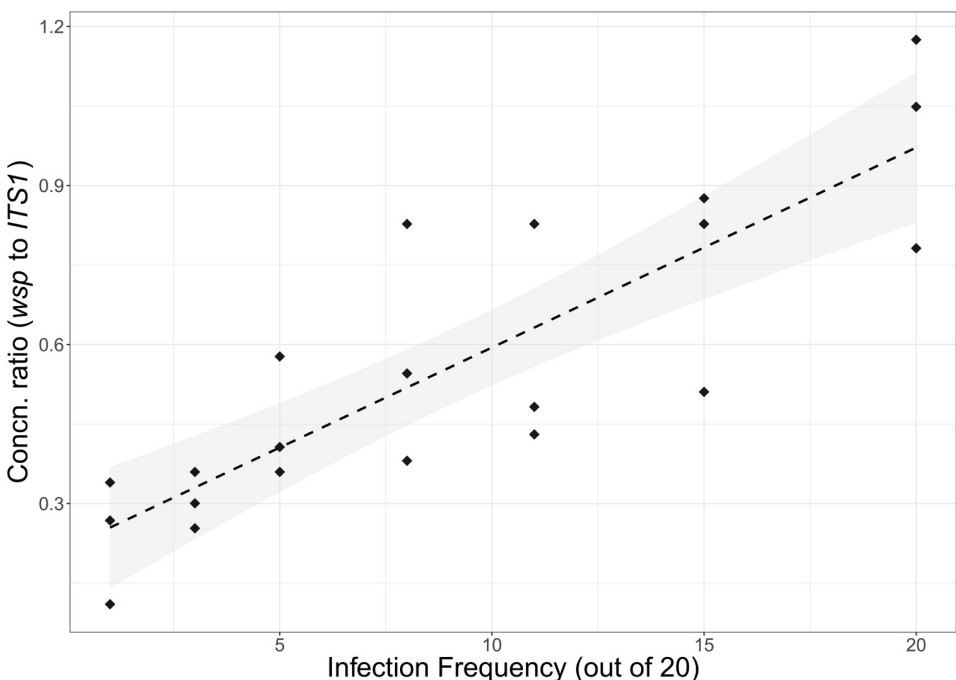

**Fig 2. Ratio of measured *w*AlbB concentrations to *Ae. aegypti* concentrations, compared to the actual infection frequency for pools of 20 mosquitoes with differing numbers of *w*AlbB-infected.** The linear regression line is shown with 95% confidence intervals shaded.

extended periods (common in adult traps) or where they have been floating in water for some time (common in ovitraps).

## Comparison with established PCR surveillance methods on Malaysian field samples

Over a series of LAMP processes, 24 individuals from each of three Malaysian locations (blindly scored as A, B and C) were tested by both LAMP and LightCycler for presence/absence of *Wolbachia*. For the LAMP results, 22/24 from location A were found to be infected (individuals 4 and 17 absent), and 22/24 from B (individuals 4 and 20 absent). All individuals of location C–which turned out to be a control site where no *Wolbachia* releases have occurred–were scored as uninfected, failing to amplify under LAMP conditions. All results were consistent with those of the qPCR assays (Table 1).

When tested with the Genie® III system, the LAMP assays developed in this paper provide a rapid, accurate and robust means of ascertaining *w*AlbB infection status of mosquitoes in field locations without the necessity for complex preparation or the use of developed laboratory facilities. LAMP assays can ascertain *w*AlbB presence/absence in pooled or individual *Ae. aegypti* mosquitoes in under 20 minutes' amplification time under a wide variety of environmental or storage conditions, and they can detect infections of 1% or lower in pooled samples, making them suitable for a variety of applications such as rapid monitoring of areas peripheral to a local *Wolbachia* release when monitoring infection spread. Applied to individuals, this *w*AlbB assay shows a similar detection sensitivity and specificity as established qPCR monitoring.

While not sufficiently precise to determine the exact frequency of *Wolbachia* in a sample, this assay when coupled with the *Ae. aegypti ITS1* primer set enables approximate

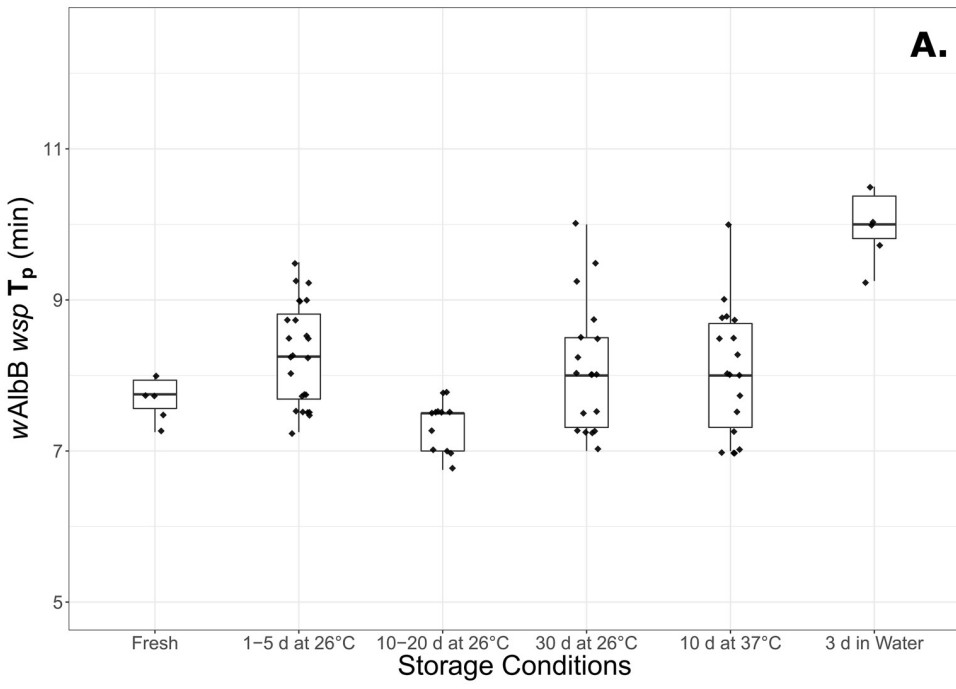

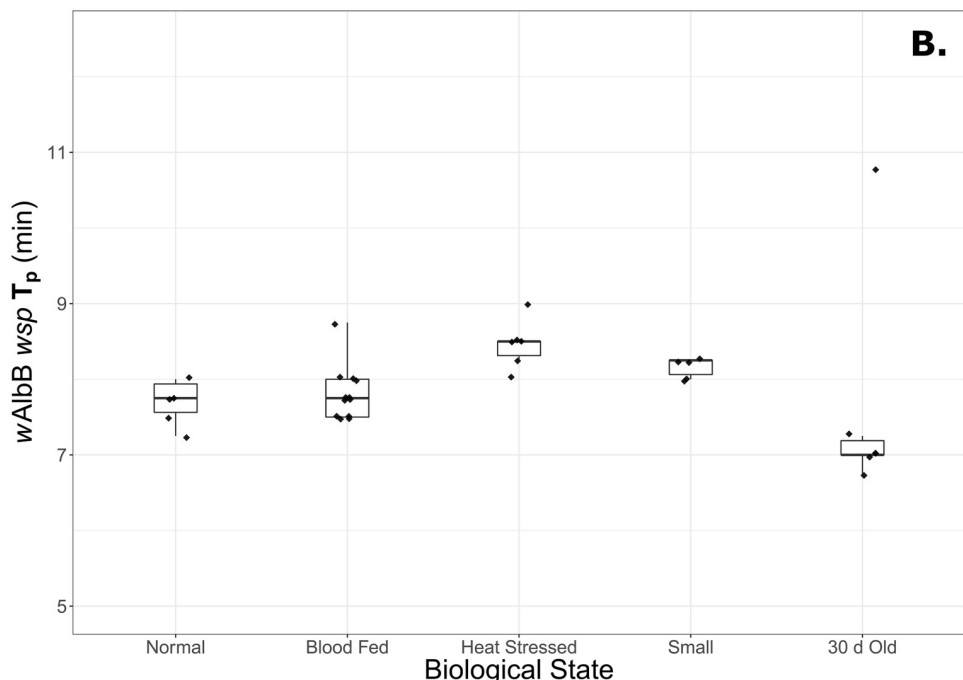

**Fig 3. Box plots of *w*AlbB peak amplification times (T$_p$) under differing storage conditions or biological states.**
**(A)** Fresh samples compared to varying lengths of dry storage at 26°C, 10 days at 37°C, and 3 days in water. **(B)** Mosquitoes under standard rearing conditions compared to mosquitoes after blood feeding, those undergoing life histories of heat or resource stress, and aged mosquitoes.

**Table 1. Relative performance of qPCR and LAMP assays for *w*AlbB detection in mosquitoes sourced from three sites in Malaysia.**

| Sample | N | Status | Infected (qPCR) | Infected (LAMP) | % difference |
|---|---|---|---|---|---|
| A | 24 | *w*AlbB release | 22 | 22 | 0 |
| B | 24 | *w*AlbB release | 22 | 22 | 0 |
| C | 24 | No release | 0 | 0 | 0 |

quantification of infection frequency within a sample. This has clear applications for monitoring *Wolbachia* deployment for disease control. While an evaluation of release success requires an estimate of *Wolbachia* frequency, approximate estimates may be adequate for many purposes when combined with selective use of more accurate assays. *Wolbachia* could first be characterized as being absent, or as having a low, intermediate or high frequency at a site using a mobile, fast-acting, method as described here. Where an approximate determination of *Wolbachia* frequencies suggest ongoing issues with a release (e.g. where frequencies remain intermediate despite ongoing releases), other more accurate estimates could be obtained. Such a two-tiered monitoring scheme would assist with the scaling of *Wolbachia* releases during an operational phase.

The potential of quantitative LAMP for monitoring and control operations remains to be fully exploited. Until this point, most qLAMP assays have focused on ascertaining the concentrations of pathogens within specific infection contexts (e.g. 14, 21, 30, 31), although studies have considered microorganismal eDNA [14], pathogenic viruses in water [30] and forensic applications [31]. Our study reveals another application–ascertaining the approximate frequency of individuals infected with a symbiotic bacterium in disease-related releases. Other frequency-based applications could include monitoring the spread of disease resistance alleles, infections, and other traits distinguishable by DNA sequence on large scales using pooled DNA. These applications are further supported by the low cost and fast run-times of the assay. Our typical expense for a single individual extracted in 200μL 0.3M KOH run as part of an 8-strip LAMP reaction with the 6-primer *w*AlbB assay is approximately $3.30 AUD per sample (for comparison, our qPCR assay for *Wolbachia* in *Ae. aegypti* is approximately $1.04 per sample—$3.12 with three replicates). Additionally, with the use of the Genie® III machine and 5-minute KOH boil in a field context, time from commencing DNA extraction to determination of *Wolbachia* status can realistically be as short as an hour. The highly specific nature of a LAMP assay, owing to its four to six primers required, means it can be easily adapted to diverse targets, provided the primer design is conducted according to best practice.

## Supporting information

**S1 Table. LAMP primers used in this study. Top.** *Wolbachia w*AlbB *wsp* LAMP assay (6-primer) [17]. **Bottom.** *Aedes aegypti ITS1* LAMP assay (6-primer) [24]. (DOCX)

**S1 Fig. Concentration curves for (A) *Ae. aegypti ITS1* and (B) *w*AlbB *wsp* each with all 6 primers.** Horizontal axes show the natural log of overall extracted DNA in the samples, while vertical axes show the peak amplification time ($T_p$) for each primer set under fresh reagent conditions. Linear regression lines are shown with 95% confidence intervals shaded. A regression of 6-primer *Ae. aegypti ITS1* $T_p$ against the natural logarithm of overall DNA concentration was found to be highly significant (p<2e-16) with an adjusted $R^2$ of 0.919 and a regression coefficient of -0.580 (S.E. 0.029). The regression of 6-primer *w*AlbB *wsp* $T_p$ against the natural logarithm of overall DNA concentration was also highly significant (p = 1.27e-13), with an adjusted $R^2$ of 0.800 and a regression coefficient of -0.423 (S.E. 0.036). The reduced $R^2$

value of *w*AlbB relative to *Ae. aegypti* may partly reflect variable concentrations of *w*AlbB within the three mosquitoes used for points along the curve.
(TIF)

## Acknowledgments

Malaysian *Ae. aegypti* samples were supplied by Dr. Nazni W. Ahmad of the Institute for Medical Research, a division of the Ministry of Health, Malaysia.

## Author Contributions

**Conceptualization:** Nicholas Bell.

**Formal analysis:** Moshe E. Jasper, Qiong Yang, Nancy Endersby-Harshman.

**Funding acquisition:** Nicholas Bell, Ary A. Hoffmann.

**Investigation:** Moshe E. Jasper, Qiong Yang, Perran A. Ross.

**Methodology:** Moshe E. Jasper, Perran A. Ross, Nancy Endersby-Harshman.

**Project administration:** Nicholas Bell.

**Resources:** Perran A. Ross, Nancy Endersby-Harshman, Nicholas Bell.

**Supervision:** Qiong Yang, Ary A. Hoffmann.

**Visualization:** Moshe E. Jasper.

**Writing – original draft:** Moshe E. Jasper.

**Writing – review & editing:** Qiong Yang, Perran A. Ross, Nancy Endersby-Harshman, Nicholas Bell, Ary A. Hoffmann.

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
