## [Decision Letter · Decision Letter 0]

10 Sep 2019

PONE-D-19-23307

A LAMP assay for the rapid and robust assessment of Wolbachia infection in Aedes aegypti under field and laboratory conditions

PLOS ONE

Dear Dr. Jasper,

Thank you for submitting your manuscript to PLOS ONE. After careful consideration, we feel that it has merit but does not fully meet PLOS ONE’s publication criteria as it currently stands. Therefore, we invite you to submit a revised version of the manuscript that addresses the points raised during the review process.

Reviewers think that after minor modifications your manuscript might be able to be accepted. Please try to respond to all queries raised by the reviewers. 

We would appreciate receiving your revised manuscript by October 15th. To enhance the reproducibility of your results, we recommend that if applicable you deposit your laboratory protocols in protocols.io, where a protocol can be assigned its own identifier (DOI) such that it can be cited independently in the future. For instructions see: http://journals.plos.org/plosone/s/submission-guidelines#loc-laboratory-protocols

We look forward to receiving your revised manuscript.

Kind regards,

Luciano Andrade Moreira, PhD

Academic Editor

PLOS ONE

Journal Requirements:

Reviewers' comments:

Reviewer's Responses to Questions

**Comments to the Author**

1. Is the manuscript technically sound, and do the data support the conclusions?

Reviewer #1: Partly

Reviewer #2: Yes

2. Has the statistical analysis been performed appropriately and rigorously? 

Reviewer #1: N/A

Reviewer #2: Yes

3. Have the authors made all data underlying the findings in their manuscript fully available?

Reviewer #1: Yes

Reviewer #2: Yes

4. Is the manuscript presented in an intelligible fashion and written in standard English?

Reviewer #1: Yes

Reviewer #2: Yes

5. Review Comments to the Author

Reviewer #1: Jasper et al. describes the optimisation and validation of a LAMP assay for wAlbB and Aedes aegypti, using primers previously published. The assay was able to detect both targets in samples from laboratory and the field, even in some simulated extreme conditions. The present study brings new findings regarding the ability to detect Wolbachia but I have some comments that must be considered, as follow:

1) The number of mosquitoes tested in each of the extreme conditions (i.e. 30 d old vs 7d old; immediately or 24hs after bloodmeal) was really low, being only 6 (Lane 137). Why was it not tested in a larger number of samples? Those extreme conditions that may happen in the field are so hard to reproduce in the lab. I would suggest to test more mosquitoes.

2) In methods, for different storage conditions, the number of mosquitoes tested should be mentioned. (lane 137-142).

3) The description of figures 1A, 1B, S1A and S1B are confusing along the manuscript. The use of two different LAMP reactions (5 or 6 primers) should be better described in methods and results and discussion at “primer validation and characterization”. It is not clear if supplementary figures were used with 5 or 6 primers. This information should be added to the legend of each figure to make it easier to be interpreted.

4) Are there figures to confirm the findings presented in lanes 235-239 (the results when 1 infected mosquito was tested with 99 uninfected and also the absence of amplification over 30min of reaction)? If so, the figure should be mentioned or I would recommend to say the data was not shown.

Also, the use of a pool of 99 uninfected mosquitoes with 1 infected should be better addressed along the manuscript.

5) The data obtained when performing the qLAMP with pooled DNA was too variable within the groups. i.e. when you have 8 infected samples, some of the pools have similar concentration when you have 12 or 15 infected out of 20. The number of samples tested was very low (n= 3), so I believe this big variation might be due to it. I would suggest running with a bigger number of samples.

Would you consider using pools in large scale releases for Wolbachia monitoring? How would you discriminate if you have 8 or 15 mosquitoes infected in a pool? An accurate method to calculate Wolbachia frequency is essential to plan releases in order to reach a population 100% Wolb-infected. I would suggest to mention this limitation in the discussion.

6)In the discussion, the LAMP assay optimised in this study was considered low cost and fast (lane 332). What is the cost per sample if you run both genes by qPCR?

Reviewer #2: This study describes the validation of two previously published sets of LAMP primers, one specific to wAlbB (and wPip) Wolbachia wsp gene and the other specific for Aedes aegypti ITS gene. While the authors did not modify the published ITS primer set, they modified the published Wolbachia assay by not using the probe that was used in the original paper for accurate visual ‘yes/no’ assay readout. Instead the authors designed a second loop primer to bind to the loop region used by the probe in order to speed up the LAMP reaction. In very well thought out experiments, the authors then validated performance characteristics of the primer sets with mosquitoes that were reared and stored in the lab under different conditions designed to mimic scenarios that would likely be encountered with field collected mosquitoes. They also directly compared performance with qPCR using field collected mosquitoes. This work would be of value to the research community because such independent validation of LAMP assays reported in literature provides necessary confirmation and confidence that these assays would indeed be successful diagnostic tools for vector surveillance under a variety of application conditions and variables.

The paper is well written and documented, however a few questions should be addressed:

1. The authors state that their aim was to improve the previously reported Wolbachia assay and that they do so by removing the sequence-specific probe used in the original assay and replacing it with a second loop primer, which as expected increases the speed of the reaction by ~5-10 min. It is debatable that this change can be considered an improvement. While assay speed was modestly increased, readout mode was changed from a sequence-specific signal (hallmark of most gold standard diagnostic applications, for instance TaqMan qPCR) to a non-sequence-specific signaling method. Moreover, this change results in an overall increase in assay cost probably due to the use of proprietary LAMP master mixes and a precision Genie II machine that costs close to $18,000. The authors should refer to their alteration of the Wolbachia assay only as a modification and not as an improvement.

2. Although the authors mention that they are using previously published LAMP assays either directly (ITS) or with a slight modification of adding one loop primer (wsp), their statements later on in the manuscript, such as “The LAMP primer set we have developed” in line 244 and “The LAMP assay we have developed” in line 290 appear misleading. These statements should be modified appropriately to indicate that in the current work the authors have validated previously published assays and made modifications that can increase speed, albeit at the cost of eliminating sequence-specific visual readout.

3. In Figure 1 the authors should indicate how many samples were tested for each DNA concentration. It appears that although there is a general trend between decreasing template amount and increasing Tp, at many positions in the standard curves lower template concentrations are being amplified faster than higher template concentrations. The authors should address how this lack of exact correlation between template amount and Tp, which is quite common for continuous amplification methods such as LAMP, would affect the accuracy of any qLAMP-based quantitations.

4. One of the most interesting notions in the manuscript was the attempt to standardize determination of Wolbachia infection frequency in a population of mosquitoes by comparing concentrations of mosquito-specific marker (ITS) with that of the Wolbachia marker. However, as shown in Figure 2, there is considerable variation. This might partly be due to the inherent inaccuracy in qLAMP quantitation. It could also be, as the authors suggest, individual to individual variation. The authors should test more mosquitoes in each of their low, mid, and high infection groups to see if this variation can be reduced to achieve more accurate prediction.

6. PLOS authors have the option to publish the peer review history of their article (what does this mean?). If published, this will include your full peer review and any attached files.

Reviewer #1: No

Reviewer #2: No

---

## [Author Response · Author response to Decision Letter 0]

29 Sep 2019

Response to reviewers

Reviewer 1.

1. Too few samples

We have increased the sample sizes for individuals in the 30 days at 26C and the 10 days at 37C categories (our most extreme, preservation-wise) to 18 to fill the gap. (See updated Figure 3). 

2. List number mosquitoes tested in methods 

These have been added (Lines 143-144)

3. Clarify 5 and 6 primer LAMP usage

Methods and results have been updated to reference construction of 5 and 6-primer sets (Lines 184-187, 236). 

Figure S1 has also been updated to clearly communicate the use of 6 primers. 

4. Better address use of pool of 99 uninfected mosquitoes

We do not present these results in figures because the only data to present are amplification times which are shown in the text.

 ‘Along the manuscript’, Lines 215-217 were updated to more clearly describe the motivation for the 99 pooled uninfected experiment. Lines 242-245 were also updated to more clearly distinguish the pooled sample experiment from the other experiments described in that section. 

5. qLAMP with pooled DNA is imprecise

We thank the reviewer for their constructive feedback. It is likely low sample sizes contribute to the variable nature of the quantitation, but we also appreciate there are limits of detection resolution. Nevertheless we believe pooled detection methods are a viable strategy for monitoring large-scale Wolbachia releases—if carefully deployed as a component of an integrated monitoring scheme. Specifically we propose a two-tier system where LAMP quantitations are used in-field to rapidly quantify approximate frequencies across a large number of sites, with a central reference lab performing qPCR to follow up on key sites as identified through LAMP results. Rolling out of this process would benefit from an in-principle field demonstration and further standardisation and calibration. 

We have updated our discussion to further acknowledge the quantitative imprecision of this pooled method, and to more extensively discuss how it could be used as part of a monitoring scheme (lines 330-341). 

6. cost of qPCR needs to be added

The cost has been added at lines 353-354

Reviewer 2.

1. Inappropriate use of the language of “improvement” on previous assay

We thank the reviewer for drawing our attention to issues around the language used to describe what is novel about our work with the previously published (Bhadra) primers & assay process. 

The 5-10 minute difference in amplification time assumes the use of our reagents & the Genie-III machine, which are integral to our assay. The original Bhadra assay calls for an amplification time of 90 minutes (compare under 25 minutes for ours). The trade-offs in terms of sequence specificity can be considered minimal for most purposes – the lack of a TaqMan-like probe is balanced by data output by Genie-III about amplification time and the annealing temperature of the product. This can be used to distinguish true amplifications from spurious amplifications – adequate for the majority of routine field monitoring uses. 

Furthermore, our assay exceeds benchmarks of the Bhadra assay in some areas. Most importantly there is a substantial improvement in the ability to detect Wolbachia in mosquitoes stored at higher temperatures (such as in BG traps which are often only checked after several days). Bhadra et. al. detected only 40% of Wolbachia infected mosquitoes stored at 4C (and none after week 2), and zero detection for samples stored at 37C. We detect Wolbachia in 100% of individuals stored at 26C for 1 month, and stored at 37C for 10 days. And we detected wAlbB in 100% of mosquitoes left in water for 3 days prior to extraction. 

As the reviewer has suggested, there will likely be many applications where the Bhadra assay remains more useful. Thus we agree that unqualified language of ‘improvement’ may not be appropriate. However, as we have outlined, our modifications do exceed the original assay on many metrics that are significant in field release monitoring. We have altered the reference to improving the Bhadra assay to instead reference ‘adapting and extending’ (line 105). We have also included some discussion explicitly comparing the two assays (lines 292-294). 

2. “we have developed” neglects dependence on previous study

To clarify the intention behind these phrases, we have altered “The LAMP primer set we have developed” to read “The modified LAMP primer set we have developed” (lines 252-253), and altered “The LAMP assay we have developed” to read, “The modified LAMP assay we have developed (lines 298-299).

3. Scatter in curve and lack of exact correlation

We provide information about the samples used for concentration curves in the methods (lines 200-207). We also quantify the strength of correlation through the statistical regression results reported in lines 271-277, as well as through the 95% confidence intervals included in the Figure 1 graph. 

We include further discussion about the imprecision issue in connection with the pooled DNA quantification (Reviewer 1 Q. 5, Reviewer 2 Q. 4). 

4. Infection frequency determination is imprecise

We agree that there is some imprecision associated with frequency determination.. Per our response to reviewer #1’s 5th comment, we have acknowledged this in our discussion (lines 330-341).

---

## [Decision Letter · Decision Letter 1]

4 Nov 2019

A LAMP assay for the rapid and robust assessment of Wolbachia infection in Aedes aegypti under field and laboratory conditions

PONE-D-19-23307R1

Dear Dr. Jasper,

We are pleased to inform you that your manuscript has been judged scientifically suitable for publication and will be formally accepted for publication once it complies with all outstanding technical requirements.

With kind regards,

Luciano Andrade Moreira, PhD

Academic Editor

PLOS ONE

Additional Editor Comments (optional):

Reviewers' comments:

Reviewer's Responses to Questions

**Comments to the Author**

1. If the authors have adequately addressed your comments raised in a previous round of review and you feel that this manuscript is now acceptable for publication, you may indicate that here to bypass the “Comments to the Author” section, enter your conflict of interest statement in the “Confidential to Editor” section, and submit your "Accept" recommendation.

Reviewer #1: All comments have been addressed

Reviewer #2: All comments have been addressed

2. Is the manuscript technically sound, and do the data support the conclusions?

Reviewer #1: Yes

Reviewer #2: Yes

3. Has the statistical analysis been performed appropriately and rigorously? 

Reviewer #1: N/A

Reviewer #2: Yes

4. Have the authors made all data underlying the findings in their manuscript fully available?

Reviewer #1: Yes

Reviewer #2: Yes

5. Is the manuscript presented in an intelligible fashion and written in standard English?

Reviewer #1: Yes

Reviewer #2: Yes

6. Review Comments to the Author

Reviewer #1: (No Response)

Reviewer #2: (No Response)

7. PLOS authors have the option to publish the peer review history of their article (what does this mean?). If published, this will include your full peer review and any attached files.

Reviewer #1: Yes: Daniela da Silva Goncalves

Reviewer #2: No

---

## [Editor Report · Acceptance letter]

12 Nov 2019

PONE-D-19-23307R1 

A LAMP assay for the rapid and robust assessment of Wolbachia infection in Aedes aegypti under field and laboratory conditions 

Dear Dr. Jasper:

I am pleased to inform you that your manuscript has been deemed suitable for publication in PLOS ONE. Congratulations! Your manuscript is now with our production department. 

With kind regards,

on behalf of

Dr. Luciano Andrade Moreira 

Academic Editor

PLOS ONE